# Validation of a Short Questionnaire to Assess Healthcare Professionals’ Perceptions of Asynchronous Telemedicine Services: The Catalan Version of the Health Optimum Telemedicine Acceptance Questionnaire

**DOI:** 10.3390/ijerph17072202

**Published:** 2020-03-25

**Authors:** Josep Vidal-Alaball, Gemma Flores Mateo, Josep Lluís Garcia Domingo, Xavier Marín Gomez, Glòria Sauch Valmaña, Anna Ruiz-Comellas, Francesc López Seguí, Francesc García Cuyàs

**Affiliations:** 1Health Promotion in Rural Areas Research Group, Gerència Territorial de la Catalunya Central, Institut Català de la Salut, 08272 Sant Fruitós de Bages, Spain; xmarin.cc.ics@gencat.cat (X.M.G.); gsauch.cc.ics@gencat.cat (G.S.V.); aruiz.cc.ics@gencat.cat (A.R.-C.); 2Unitat de Suport a la Recerca de la Catalunya Central, Fundació Institut Universitari per a la Recerca a l’Atenció Primària de Salut Jordi Gol i Gurina, 08272 Sant Fruitós de Bages, Spain; 3Department of Economics and Business, Universitat de Vic-Universitat Central de Catalunya, 08500 Vic, Spain; jlgarcia@uvic.cat; 4Unitat d’anàlisi i Qualitat, Xarxa Sanitària i Social de Santa Tecla, 43003 Tarragona, Spain; 5Centre d’Atenció Primària Sant Joan de Vilatorrada, Gerència Territorial de la Catalunya Central, Institut Català de la Salut, 08250 Sant Joan de Vilatorrada, Spain; 6TIC Salut Social–Generalitat de Catalunya, 08005 Barcelona, Spain; flopez@ticsalutsocial.cat; 7CRES&CEXS–Universitat Pompeu Fabra, 08002 Barcelona, Spain; 8Hospital Sant Joan de Déu, Digital Care Research Group, Universitat de Vic-Universitat Central de Catalunya, 08500 Vic, Spain; francesc.garcia@umedicina.cat

**Keywords:** telemedicine, questionnaires and surveys, validation studies, health personnel

## Abstract

Telemedicine is both effective and able to provide efficient care at a lower cost. It also enjoys a high degree of acceptance among users. The Technology Acceptance Model proposed is based on the two main concepts of ease of use and perceived usefulness and is comprised of three dimensions: the individual context, the technological context and the implementation or organizational context. At present, no short, validated questionnaire exists in Catalonia to evaluate the acceptance of telemedicine services amongst healthcare professionals using a technology acceptance model. This article aims to statistically validate the Catalan version of the EU project Health Optimum telemedicine acceptance questionnaire. The study included the following phases: adaptation and translation of the questionnaire into Catalan and psychometric validation with construct (exploratory factor analysis), consistency (Cronbach’s alpha) and stability (test–retest) analysis. After deleting incomplete responses, calculations were made using 33 participants. The internal consistency measured with the Cronbach’s alpha coefficient was good with an alpha coefficient of 0.84 (95%, CI: 0.79–0.84). The intraclass correlation coefficient was 0.93 (95% CI: 0.852–0.964). The Kaiser–Meyer–Olkin test of sampling showed to be adequate (KMO = 0.818) and the Bartlett test of sphericity was significant (Chi-square 424.188; gl = 28; *p* < 0.001). The questionnaire had two dimensions which accounted for 61.2% of the total variance: quality and technical difficulties relating to telemedicine. The findings of this study suggest that the validated questionnaire has robust statistical features that make it a good predictive model of healthcare professional’s satisfaction with telemedicine programs.

## 1. Introduction

Although at present there is no clear consensus as to the economic impact of telemedicine [1,2], recent evidence has shown it can be effective, provide efficient care at a lower cost [3,4,5,6,7] and enjoy a high degree of acceptance among users. Moreover, telemedicine reduces journeys by road and therefore decreases the environmental impact of atmospheric pollutants emitted by vehicles [8]. Published studies have already provided some insights as to the acceptance drivers. Eddy et al. reported high patient satisfaction with teledermatology and amongst physicians, although this satisfaction was higher in primary care doctors than in dermatologists [9]. McKoy et al. used questionnaires to assess the acceptance of a teledermatology service and reported that 82% of users saw it as a valid alternative to face-to-face consultations [10]. In another qualitative study using semi-structured interviews with 32 healthcare professionals, MacNeill et al. showed mixed points of view: while it was broadly welcomed by nursing staff, some primary care physicians were worried that telemedicine could increase their workload and it could potentially undermine their professional autonomy [11]. A comprehensive systematic review recently published by Mounessa et al. reported that patients and healthcare providers were in general highly satisfied with the two types of telemedicine: store-and-forward and real time telemedicine [11]. Whilst all the studies provided valuable inputs to help understand the complex heterogeneous effect of the participants’ acceptance of new healthcare models, it is of the utmost importance that any published evidence uses validated questionnaires to perform reliable and comprehensive evaluations [12].

The Technology Acceptance Model (TAM) proposed by Davis (1989) [11] is based on the two main concepts of ease of use and perceived usefulness, and comprises three dimensions: the individual context, the technological context and the implementation or organizational context. Based on his benchmark, studies have adapted their methodology to create validated questionnaires. For example, Orruño et al. (2011) evaluated teledermatology adoption by healthcare professionals using a modified 33-item version of this model grouped into eight theoretical dimensions. The Cronbach’s alpha for each theoretical variable and internal consistency of the questionnaire reported good results which suggest the proposed questionnaire is a valid tool for assessing acceptance in this setting [13].

In the REgioNs of Europe WorkINg toGether for HEALTH (RENEWING HEALTH) project, Kidholm et al. reviewed the scientific literature to find questionnaires used in European telemedicine projects to assess the stakeholders’ perceptions. One such questionnaire was used in the EU project Health Optimum (Delivery OPTIMisation through telemedicine) [14,15]. This questionnaire for healthcare professionals includes eight general questions irrespective of their medical specialty and focuses on the physicians’ perception of the quality of the telemedicine service, their convenience, technical and other difficulties and potential effects on the health of the patients using the service (see Appendix A). The questionnaire was not validated and although it does not strictly use the TAM, it adopts some of its dimensions. An easy-to-answer questionnaire increases the response rate but no other short validated questionnaire using the TAM to assess healthcare professionals’ perceptions of asynchronous telemedicine services was found.

Whilst the Catalan Ministry of Health is putting a lot of effort into developing telemedicine, the degree of acceptance of these tools has yet to be investigated using validated questionnaires.

It is not possible to deploy telehealth services without first carrying out a thorough assessment of practitioner’s perceptions of their usefulness, a factor which is critical in fostering their deployment in public and private healthcare systems. Brief questionnaires are preferable as a means to improve response rates as studies using long questionnaires based on the TAM have reported low response rates [16,17]. For these reasons, the aim of our study is the statistical validation of the Catalan version of the EU project Health Optimum telemedicine acceptance questionnaire.

## 2. Methods

### 2.1. Likert Scale

The original questionnaire used an incomplete Likert scale with an even number of answers to the questions, while it is recommended that questionnaires use an odd number of answers [18]. Furthermore, there were more positive options than negative in the first two questions. We decided to add an extra possible response to the first four questions in order to have an odd number of options and therefore obtain a complete Likert scale.

### 2.2. Translation into Catalan and Data Collection

To achieve the higher content validity, a translated and back-translated methodology was used [19,20]. The original English version of the questionnaire (Appendix A) was translated into Catalan independently by two authors who are native Catalan speakers and who are both proficient in English. An agreed Catalan version of the questionnaire was obtained following several drafts (Appendix B).

In order to validate the new Catalan version of the Health Optimum questionnaire, the Google Forms tool was used to send it to primary healthcare professionals in the Catalan central region who had used telemedicine services in the past. WhatsApp health professional groups were used to disseminate the questionnaire. Members were asked to answer the questionnaire and to resend it to other potential respondents. The twitter account of the principal investigator (@jvalaball, >10K followers) was also used for further dissemination. Additional information regarding the respondents’ basic characteristics was added to the questionnaire: age, sex and professional role and the kind of telemedicine services available in the Catalan central region which they had used (teledermatology, teleulcers or teleaudiometries). Non health professionals were asked not to answer the questionnaire.

The questionnaire called “Questionnaire to assess healthcare professionals’ perceptions of asynchronous telemedicine services” (Qüestionari per avaluar la percepció dels professionals sanitaris amb els serveis de telemedicina asíncrona) had 8 questions with a complete Likert scale of 5 answers the first 4 questions and 3 answers the last 4 questions. It was first sent at the beginning of April 2018 and it was closed 3 d later. It was resent 2 weeks later, asking participants to answer it again in order to check for consistency. The questionnaire was closed definitively 5 d later. Completing the questionnaire was considered as an indication of consent to participate in the study. The study protocol was approved by the University Institute for Primary Care Research (IDIAP) Jordi Gol Health Care Ethics Committee (Code P16/046).

### 2.3. Scale Level Descriptive Analysis

Following Argimon et al.’s methodology [21], we have assessed the variability in responses to the questionnaire calculating the average and the standard deviation (SD) and calculated the frequencies to check for floor and ceiling effects [22]. These effects are important as they can influence the validity, reliability and responsiveness of a questionnaire and they are used to check the percentage of participants with very low and very high scores. We have taken this effect to exist when 15% or more of the responses are found in the higher or lower values [23].

We checked the discriminating capacity of the items using the discriminative rate, which compares the responses in the two extreme groups (individuals who have obtained a total score below the 33rd percentile and individuals who have scored above the 66th percentile). Discriminative rates above 0 indicate discriminating capacity of the items [22].

### 2.4. Internal Consistency

The internal consistency of the Catalan version of the Health Optimum questionnaire was tested using Cronbach’s alpha. This statistical test checks for the degree of common information that share the items of a scale of measurement. It is an average of the variances between the variables that are part of a scale. Values of α above 0.7 indicate an appropriate internal consistency [22].

### 2.5. Temporal Stability—Reliability

To check for the intra-observer stability of our questionnaire, a test–retest methodology was used. Its reliability was calculated with intraclass correlations (ICC) between the scores at Time 1 and Time 2 at the individual item level. Intraclass correlation coefficients over 0.75 are considered good and over 0.90 excellent [22].

### 2.6. Factor Analysis 

Factor analysis was used to simplify the information given by a correlation matrix to make it more easily interpretable [23]. To check for the appropriate conditions to perform a factor analysis we have used 2 methods: the Kaiser–Meyer–Olkin test and the Bartlett sphericity test. The first ranges from 0 to 1, with a value lower than 0.5 indicating that it is inappropriate to do the analysis. A significant result of the Bartlett sphericity test (*p* < 0.05) indicates that it is pertinent to make a factor analysis. To check for the number of different dimensions of the questionnaire, 3 different criteria were used: (1) Kaiser rule which selects the number of factors with a value greater than 1; (2) the percentage of explained variance which is determined by the accumulated percentage of variation extracted in each factor (varimax rotation with Kaiser’s normalization was used to simplify the number of dimensions); (3) a scree plot which graphically represents the number of factors or dimensions extracted, we retained the factors or components to the left of the inflection point on the graph [23].

The statistical programs STATA version 15/SE and SPSS v23 (SPSS Inc., Chicago, IL, USA) were used for these statistical analyses. Results were considered significant with *p* < 0.05.

## 3. Results

### 3.1. Test–Retest

Although 212 responses were received, only 37 individuals responded to the questionnaire twice as required. After checking the response times, we found four respondents that although responded twice to the questionnaire, didn’t wait the necessary minimum two weeks period between the responses. After excluding these other four respondents, calculations were made using 33 participants: 24/33 were women (72%) and their average age was 50.9 (SD: 8.87) years. Among them, 24/33 (72%) were family physicians, 2/33 were nurses (6%), 1/33 was a dermatologist (3%) and 4/33 (12%) had other specialties.

The internal consistency measured with the Cronbach’s alpha coefficient showed none of the items significantly altered the consistency of the instrument. The overall alpha coefficient was 0.84 (95%, CI: 0.79–0.84). Table 1 shows the alpha coefficient for each of the eight items in the questionnaire. With respect to temporal stability, the intraclass correlation coefficient value stands at 0.93 (95% CI: 0.852–0.964), thus showing excellent reliability of the test.

### 3.2. Descriptive Analysis of the Items

All discriminative rates were above zero and, for the first four questions, they were above one. The highest discrimination item was Item 3, “Clinical quality” (discrimination index = 1.40), and the lowest discrimination item was Item 8, “Future use” (0.51). Table 1 describes the average score for each group and the discrimination index of each of the eight items in the questionnaire.

The lowest ceiling effect score was for Item 3 “Clinical quality” and the highest score was for Item 5, “Health effects”. Furthermore, two out of five items were above the ceiling-effect criterion of 15%.

### 3.3. Exploratory Factor Analysis 

The Kaiser–Meyer–Olkin test of sampling was adequate (KMO = 0.818) and the Bartlett test of sphericity was significant (Chi-square 424.188; gl = 28; *p* < 0.001), indicating that the items were appropriate for a factor analysis. Two factors emerged with an eigenvalue greater than one (Table 2). The factor with questions about the quality of telemedicine technology (Items 1–5) was named Quality. The other factor contained items relating to technical difficulties in telemedicine (Items 6–8) and was named Difficulties. The two constructs together accounted for 61.2% of total variance, all factor loadings being higher than 0.40. Figure 1 shows the scree plot representing the number of dimensions extracted.

These results show that the Catalan version of the Health Optimum questionnaire to assess practitioner’s perceptions of telemedicine tools is statistically robust.

## 4. Discussion

The validation of the telemedicine questionnaire for healthcare professionals showed good reliability and an acceptable level of validity. Two questions suffered from the ceiling effect which limit the validity, but all the questions had discriminative rates above zero, higher for the first four questions, showing that all the questions had a discriminative capacity. We could increase the response options to the items to help optimize their discriminative power, but this would reduce the comparability of the scores with the original questionnaire.

The internal consistency was good, with Cronbach’s alpha coefficients for the overall questionnaire and Factor 1 above 0.80. The Cronbach’s alpha coefficient was slightly below the acceptable cut-off for Factor 2 (0.67). The low number of items included in the second factor may contribute to this finding. These results are slightly worse than the one reported by Argimon et al. using a similar methodology [24]. The ceiling effect was found in two questions, though this was not measured in the English original version, meaning that we are unable to conclude that it is an intrinsic characteristic of the questionnaire or a weakness in the Catalan translation.

Our results show that the Catalan version of the Health Optimum questionnaire is a robust tool for assessing healthcare professional’s satisfaction with a telemedicine program. All the questions presented a positive discrimination rate, especially Item 3, “Clinical quality”. This means that the questionnaire is clearly able to distinguish the clinical quality of a telemedicine program.

The results show that it was adequate to perform the factor analysis and showed that the questionnaire had two dimensions, which accounts for 61.2% of the total variance: one concerning the quality of the telemedicine technology and another about technical difficulties relating to telemedicine.

The questionnaire will be useful to evaluate healthcare professionals’ perceptions of the growing number of different asynchronous telemedicine programs available in Catalonia as well as allowing comparisons between them to inform which characteristics make them more accepted by professionals [25].

### Limitations

This study has several limitations. The pilot questionnaire was distributed to healthcare professionals via WhatsApp healthcare professional groups asking them to only answer if they had used a telemedicine program in the past. As the questionnaire was anonymous we can not verify whether they had in fact used these programs and we are unable to be sure that the questionnaire was not distributed to other individuals who are unrelated to healthcare professions. The number of respondents (33) was small compared to other studies [24], nevertheless this number was suficient to validate the questionnaire.

## 5. Conclusions

The Catalan version of the Health Optimum telemedicine acceptance questionnaire has been validated with this study showing robust statistical features that make it a quality tool to assess healthcare professional’s satisfaction with telemedicine programs. As the validation was not performed looking at a specific telemedicine program, the validated questionnaire is potentially valid for any telemedicine program.

## Figures and Tables

**Figure 1 ijerph-17-02202-f001:**
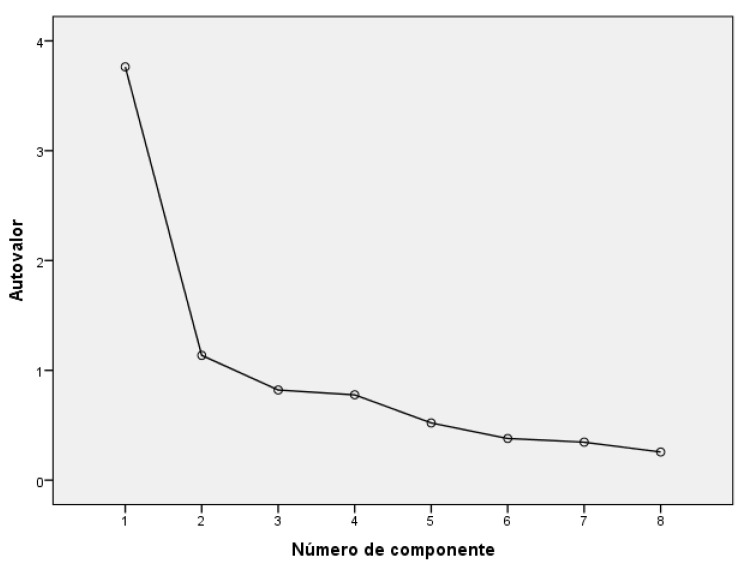
Scree plot.

**Table 1 ijerph-17-02202-t001:** Psychometric validation.

	Descriptive Statistics		Item Parameter Estimates			Reliability	
Questionnaire’s Items	Mean	Standard Deviation	IDI ^1^	Ceiling Effect (%)	Floor Effect (%)	Item-Correlation	Cronbach Alpha
1. Global quality ^a^	3.51	0.85	1.24	8.3	1.4	0.79	0.79
2. Technical quality ^a^	3.47	0.87	1.13	5.6	2.8	0.75	0.80
3. Clinical quality ^a^	3.01	0.86	1.40	2.8	0.7	0.73	0.81
4. Convenience ^a^	3.67	0.89	1.36	16.0	1.4	0.75	0.80
5. Health effects ^b^	2.68	0.61	0.54	75.7	-	0.56	0.83
6. Technical difficulties ^b^	2.03	0.56	0.59	-	-	0.67	0.81
7. Organizational difficulties ^b^	2.10	0.57	0.60	-	-	0.64	0.82
8. Future use ^b^	2.53	0.51	0.51	-	-	0.54	0.84

^1^ IDI: Item Discrimination Index, ^a^ Items scored on a 5-point response scale ranging from 1 “very dissatisfied” to 5 “very satisfied”; ^b^ Items scored on a 3-point response scale.

**Table 2 ijerph-17-02202-t002:** Exploratory factor analysis (EFA): data on commonalities of items, item loadings in Factor 1 and Factor 2.

Questionnaire Items	Factor 1	Factor 2	Communalities (*h*^2^)
1. Global quality ^a^	0.712	0.430	0.692
2. Technical quality ^a^	0.526	0.571	0.603
3. Clinical quality ^a^	0.785	0.234	0.671
4. Convenience ^a^	0.827	0.201	0.722
5. Health effects ^a^	0.670	-	0.450
6. Technical difficulties ^b^	0.154	0.836	0.722
7. Organizational difficulties ^b^	-	0.870	0.763
8. Future use ^b^	0.281	0.441	0.274
Eigenvalues	3.76	1.14	
Average variance explained (%)	47.04	14.21	
Cronbach’s α reliability	0.830	0.672	

Extraction method, principal component analysis, rotation method, varimax with Kaiser normalization; rotation converged in three iterations. Factors: ^a^ Quality, ^b^ Difficulties.

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
