# Peer review of "Validation of a Short Questionnaire to Assess Healthcare Professionals’ Perceptions of Asynchronous Telemedicine Services: The Catalan Version of the Health Optimum Telemedicine Acceptance Questionnaire"

_ijerph, 2020, doi:10.3390/ijerph17072202_

Round 1
Reviewer 1 Report
The idea of this paper is interesting, but I have a little problem with the execution. I agree that there probably is a need for validated measurement - tools for tech. acceptance - however I think this survey is very basic - and there is no explanation behind the choice of questions or relating it to some kind of theory or logic. There is no theory/ literature chapter which states the contribution of this study and discuss f.ex other surveys and their shortcoming or discuss the questions in relation to a theory (technology acceptance?) Also, the survey is very simple and there are no indexes to ensure reliability of an item (such as factors etc.). Unless, one can see the whole survey as a one factor - but then I think it should be measured in relation to a dependent variable and perhaps also test with more than one sample, or in another way show a variation.
Author Response
Thank you for your valuable comments. Note that line numbers may vary depending on your Microsoft word version
- We explained that the questions where chosen following the Technology Acceptance Model proposed by Davis (line 65). We have chosen a short questionnaire because other studies reported low response rates. We have now explained this in the manuscript and added two references to support it (lines 90-91).
Reviewer 2 Report
Authors may give examples of telemedicine to highlight its importance.
The conclusions are rather sketchy. Link the research back to telemedicine, and suggest future research areas.
Author Response
Response:
Thank you for your valuable comments. Note that line numbers may vary depending on your Microsoft word version
- We have added a couple of references as examples of the importance of telemedicine (lines 48-51):
[7] R. L. Bashshur, G. W. Shannon, T. Tejasvi, J. Kvedar, and M. Gates, “The Empirical Foundations of Teledermatology: A Review of the Research Evidence,” Telemed. e-Health, vol. 21, no. 12, pp. 10–27, 2015.
[8] J. Vidal-Alaball, J. Franch-Parella, F. Lopez Segui, F. Garcia Cuyàs, and J. Mendioroz Peña, “Impact of a Telemedicine Program on the Reduction in the Emission of Atmospheric Pollutants and Journeys by Road,” Int. J. Environ. Res. Public Heal., vol. 16, no. 22, p. 4366, 2019.
The conclusions are rather sketchy. Link the research back to telemedicine, and suggest future research areas.
Response:
- We have done this is the discussion section and included you suggestion with another one similar done by reviewer 3. As the questionnaire has been validated, in the future we will be using it to assess healthcare professionals’ perceptions of asynchronous telemedicine services in our region (lines 226-229).
Reviewer 3 Report
Authors presented manuscript about validation of a short questionnaire to assess healthcare professionals’ perceptions of asynchronous telemedicine services. As a reviewer I have some comments and suggestions due to the manuscript.
In abstract should be without headings but has the structure: Background (with the purpose of the study), Methods, Results and Conclusion.
There is different size of letter in manuscript,
At the end of first paragraph in section Introduction there should be added references at the end of this paragraph – Authors have to verify whole manuscript because there is more place where should be added references – check it one more time,
Section second Materials and Methods please provide information of inclusion and exclusion criteria of participants.
In this section should be presented name of the validated questionnaire in English and Catalan.
The information about the validated questionnaire should be presented – for example number of questions, and …
In section Results it is quite confused – how many participants took part in this research and finally how many was analyzed – and explain way there were excluded.
Section Discussion is very short – may be compare your results with original or other validated versions or present more information about telemedicine…
Why some words in text are bolt?
The References must be modified due to Instruction for Authors.
Author Response
In abstract should be without headings but have the structure: Background (with the purpose of the study), Methods, Results and Conclusion.
Response:
Thank you for your valuable comments. Note that line numbers may vary depending on your Microsoft word version
- We have removed the heading of the abstract. Lines 24 to 43.
There is different size of letter in manuscript.
Response:
- This is some type of error with Microsoft word versions; we have corrected this in all the manuscript.
At the end of first paragraph in section Introduction there should be added references at the end of this paragraph – Authors have to verify whole manuscript because there is more place where should be added references – check it one more time
Response:
- We have added a reference at the end of the paragraph (line 64):
[12] L. Thijssing, E. Tensen, and M. Jaspers, “Patient’s Perspective on Quality of Teleconsultation Services,” Stud Heal. Technol Inf., vol. 228, no. 6, pp. 132–136, 2016.
- We have checked again the manuscript correcting references and deleted a duplication of reference 15 (line 89).
Section second Materials and Methods please provide information of inclusion and exclusion criteria of participants.
Response:
- Inclusion criteria is already mentioned. We have added exclusion criteria of participants (lines 113-114).
In this section should be presented name of the validated questionnaire in English and Catalan.
Response:
- We have added the name of the questionnaire: “Questionnaire to assess healthcare professionals’ perceptions of asynchronous telemedicine services” (Qüestionari per avaluar la percepció dels professionals sanitaris amb els serveis de telemedicina asíncrona). Lines 117-119.
The information about the validated questionnaire should be presented – for example number of questions, and …
Response:
- We have added this information: the questionnaire had 8 questions with a complete Likert scale of 5 answers the first 4 questions and 3 answers the last 4 questions. Lines 119-120.
In section Results it is quite confused – how many participants took part in this research and finally how many was analyzed – and explain way there were excluded.
Response:
- Yes, you are right, it looks confusing. We already explain that finally we included 33 individuals but we have simplified the explanation in order to minimize confusion. Lines 163-165.
Section Discussion is very short – may be compare your results with original or other validated versions or present more information about telemedicine…
Response:
- We have added more information on this section. We have written about the utility of this questionnaire to compare telemedicine services (lines 227-230). We have also compared our results with the ones of an article that used a similar methodology (lines 214-215).
Why some words in text are bolt?
Response:
- This is some type of error with Microsoft word versions; we have corrected this in all the manuscript.
The References must be modified due to Instruction for Authors.
Response:
- We have emended references according the instruction for authors.